# Can Low-Dose of Dietary Vitamin E Supplementation Reduce Exercise-Induced Muscle Damage and Oxidative Stress? A Meta-Analysis of Randomized Controlled Trials

**DOI:** 10.3390/nu14081599

**Published:** 2022-04-12

**Authors:** Myunghee Kim, Hyeyoon Eo, Josephine Gahyun Lim, Hyunjung Lim, Yunsook Lim

**Affiliations:** 1Department of Food and Nutrition, Kyung Hee University, Seoul 02447, Korea; ddorothy8@naver.com (M.K.); gahyun146@khu.ac.kr (J.G.L.); 2Department of Biomedical and Pharmaceutical Sciences, Graduate School, Kyung Hee University, Seoul 02447, Korea; hyeyooneo@khu.ac.kr; 3Department of Medical Nutrition, Kyung Hee University, Yongin 17104, Korea; hjlim@khu.ac.kr

**Keywords:** exercise, muscle damage, oxidative stress, inflammation, vitamin E

## Abstract

Vitamin E plays an important role in attenuating muscle damage caused by oxidative stress and inflammation. Despites of beneficial effects from antioxidant supplementation, effects of antioxidants on exercise-induced muscle damage are still unclear. The aim of this meta-analysis was to investigate the effects of dietary vitamin E supplementation on exercise-induced muscle damage, oxidative stress, and inflammation in randomized controlled trials (RCTs). The literature search was conducted through PubMed, Medline, Science Direct, Scopus, SPORTDiscuss, EBSCO, Google Scholar database up to February 2022. A total of 44 RCTs were selected, quality was assessed according to the Cochrane collaboration risk of bias tool (CCRBT), and they were analyzed by Revman 5.3. Dietary vitamin E supplementation had a protective effect on muscle damage represented by creatine kinase (CK; SMD −1.00, 95% CI: −1.95, −0.06) and lactate dehydrogenase (SMD −1.80, 95% CI: −3.21, −0.39). Muscle damage was more reduced when CK was measured immediately after exercise (SMD −1.89, 95% CI: −3.39, −0.39) and subjects were athletes (SMD −5.15, 95% CI: −9.92, −0.39). Especially vitamin E supplementation lower than 500 IU had more beneficial effects on exercise-induced muscle damage as measured by CK (SMD −1.94, 95% CI: −2.99, −0.89). In conclusion, dietary vitamin E supplementation lower than 500 IU could prevent exercise-induced muscle damage and had greater impact on athletes

## 1. Introduction

It is well-known that exercise improves whole-body energy metabolism and makes muscles stronger and more resistant to fatigue [1]. In addition, regular exercise improves cognitive functions in healthy populations [1]. Due to the beneficial effects of exercise as mentioned, interest in physical activity is increasing all over the world. 

According to the scientific evidence reported, there are several molecules and signaling cascades involving in muscle health. Among those factors, reactive oxygen species (ROS) is generated during exercise and makes muscles stronger as well as induces adaptation by up-regulating endogenous antioxidant enzymes [2]. The enzymatic antioxidants increase resistance to fatigue, reduce oxidative stress, and ultimately enhance whole body health [3]. Exercise-induced ROS also activates redox-sensitive signal pathways that control inflammatory transcriptional factors [4]. Especially, activating inflammatory response induces expression of genes which facilitate the regeneration of damaged skeletal muscle [4]. Moreover, several studies have indicated that exercise-induced ROS and inflammation resulted in muscle adaptation such as strengthened endogenous antioxidant defense [5], mitochondrial biogenesis, and regenerative response [6].

Although exercise-induced ROS and inflammation showed advantageous effects as mentioned above, overwhelming ROS and chronic inflammation during intensive physical exercise may be harmful for skeletal muscle through damage of cellular components including proteins, lipids, and nucleic acids [7,8]. As a result of the damage, oxidative products such as protein carbonyls (PC), F2-isoprostanes, malondialdehydes (MDA) and 8-oxo-2′-deoxyguanosine (8-OHdG) [7] as well as inflammatory myokines such as interleukin 6 (IL-6) and tumor necrosis factor-α (TNF-α) are increased and consequently hinder muscular performance and delay the recovery [8]. Additionally, strenuous physical exercise, which generates excessive oxidative stress and inflammation, leads to delayed-onset muscle soreness, muscle stiffness, and muscle weakness, indicated by increased levels of creatine kinase (CK) and lactate dehydrogenase (LDH) [9].

Antioxidant and anti-inflammatory nutrients received great attention as a possible solution for preventing exercise-induced oxidative stress and inflammation [10,11,12]. Vitamin E as a lipid-soluble and chain-breaking antioxidant, stops the progression of the lipid peroxidation chain reaction and maintains the integrity of polyunsaturated fatty acids in the cell membranes [13,14]. Researchers have investigated whether dietary vitamin E supplementation prevents exercise-induced muscle damage, oxidative stress, and inflammation. For example, many studies reported that vitamin E supplementation decreased CK [15,16,17] and MDA [18,19,20] concentration following exercise. However, other studies demonstrated that vitamin E supplementation did not decrease CK and MDA levels after exercise [21,22], which is opposite to the reports mentioned above. In addition, a previous study showed that vitamin E supplementation decreased IL-6 [23], while another study showed no effect from vitamin E supplementation on IL-6 [24].

Taken together, vitamin E supplementation showed controversial effects on exercise-induced muscle damage, oxidative stress, and inflammation. Because a previous meta-analysis of six randomized controlled trials (RCTs) found no beneficial effects of vitamin E supplementation on exercise-induced lipid peroxidation and muscle damage [25], multifactorial studies are needed to investigate the overall antioxidant and anti-inflammatory effects of dietary vitamin E supplementation on exercise-induced muscle damage, oxidative stress, and inflammation. Hence, the present study investigated the effects of dietary vitamin E supplementation with subgroup analysis to consider various factors.

## 2. Materials and Methods

### 2.1. Literature Search Strategy

The literature search for a meta-analysis was performed using the following databases: PubMed, Science Direct, Scopus, SPORTDiscuss, EBSCO. The final search was carried out in February 2022. To search for vitamin E, the following search keywords were used: “antioxidant”, “vitamin E”, “tocopherol”, and “alpha tocopherol”. To search for exercise, the following search keywords were used: “exercise”, “physical activity”, “acute exercise”, “isometric exercise”, “aerobic exercise”, “anaerobic exercise”, “endurance exercise”, “running”, “treadmill”, “cycling”, “bicycle ergometry”, and “training”. To search for muscle damage, the following terms were used: “muscle damage”, “muscle soreness”, “muscle pain”, “creatine kinase (CK)”, and “lactate dehydrogenase (LDH)”. To search for exercise-induced oxidative stress, the following search keywords were used: “malondialdehyde (MDA)”, “protein carbonyls (PC)”, “8-hydroxy-2deoxyguanosine (8OHdG)”, “total radical scavenging capacity (TRSC)”, “4-hydroxynonemal (4-HNE)”, “F2-isoprostane”, “total antioxidant status (TAS)”, “glutathione peroxidase (GLP)”, and “glutathione”. To search for exercise-induced inflammation, the following terms were used: the acute phase protein C reactive protein (CRP)”, “interleukin-6 (IL-6)”, “interleukin-10 (IL-10)”, and “tumor necrosis factor- α (TNF-α)”. 

### 2.2. Study Selection

The literature which fulfilled the following inclusion and exclusion criteria was selected. The inclusion criteria were: (1) full-text study with human-based randomized clinical trials (RCTs); (2) study on healthy participants (age 18 < age < 65 years old); (3) study with only pre-exercise supplementation. The exclusion criteria were (1) meta-analyses and reviews; (2) studies using a mixture of other antioxidants for supplementation; (3) studies on smokers, children (≤18 years old), the elderly (≥65 years old), and patients; (4) studies with no exercise after supplementation. Exercise protocol which attempts to detect the changes in biomarkers after intake of final supplementation was considered valid [26]. When the results of the selected studies were only graphically represented, the authors of those studies were contacted via email. If the information was still unavailable due to no response from the author or data loss, transformation of graphical data to numerical was performed using the Digitizelt program, a digitizer measuring tool (Digitizelt 2015; Bormann, Braunschweig, Germany). A biomarker was analyzed if it was measured in two or more studies with a similar timeframe. The process of study selection was conducted according to the Preferred Reporting Items for Systematic Reviews and Meta-Analyses flow diagram (PRISMA).

### 2.3. Quality Assessment

The quality of included studies was judged according to the Cochrane collaboration risk of bias tool (CCRBT) [27]. CCRBT is composed of eight domains, which are random sequence generation, allocation concealment, blinding of participants/personnel/outcome assessor, incomplete output data, selective reporting, and other potential threats to validity. Each domain was evaluated as ‘yes’, ‘no’, and ‘unclear’, and two evaluators (M.K. and Y.L.) judged the bias according to the detailed criteria. Literature with a ‘yes’ in all domains was classified as low risk of bias. Literature with ‘unclear’ or ‘no’ in two or fewer domains was classified as moderate risk of bias. Literature with ‘unclear’ or ‘no’ in three or more domains was classified as high risk of bias.

### 2.4. Statistical Analysis

Data analysis was performed using Revman 5.3 (Cochrane Collaboration, Oxford, UK). Because there is high variation in the duration of supplementation and exercise protocol (method, intensity, duration), the random effect analysis model was applied. Data were extracted in the form of means and standard deviations (SD). The mean difference was defined as the difference in each biomarker between the placebo and supplemented groups at pre- and post-exercise. Effect size (Z) was drawn from the mean difference and was interpreted based on the following reference points: Z values of 0.2, 0.5, and 0.8 were considered small, moderate, and large, respectively [28]. The degree of inconsistency within studies, also referred to as heterogeneity (I^2^), was interpreted based on the following reference points: 25%, 50%, and 75%, representing no, low, moderate, and high heterogeneity, respectively [29]. Subgroup analyses were performed based on time points (immediately, 24 h, and 48 h after exercise), vitamin E supplementation dosage (≤500, >500 IU), and subject type (athlete, non-athlete) to specifically evaluate the effects of vitamin E supplementation on exercise-induced CK and MDA. If *p*-value was *p* ≤ 0.05, the result was considered statistically significant. To assess publication bias, Egger’s test and Funnel plot were used to explore the possibility of small-study effects (a tendency for estimates of the intervention effect to be more beneficial in smaller studies), proceeded by software R (Version 3.6.1, meta-package, 2019) and Revman, respectively. If the *p*-value of the Egger’s test was *p* < 0.05, it was considered to have publication bias.

## 3. Results

### 3.1. Literature Research

The search was performed up to February 2022 and covered all studies published in the previous years. Seven hundred and eighty-two RCTs were initially screened and repeated, and 464 RCTs were excluded. After further screening based on title and abstract, 288 RCTs were excluded because they used other antioxidants and mixtures of antioxidants for supplementation, or were not conducted in healthy people, conducted in the elderly, children, smokers, and patients, or were not full-text. For the meta-analysis, 44 studies with comparable makers, measurement frequencies, and valid exercise protocols were finally selected. Finally, 17 RCTs were selected and measured biomarkers in the studies were CK, LDH, MDA, TAS, and IL-6. Details of the selection process are represented in the flow chart (Figure 1).

### 3.2. Characteristics of the Included Studies

Total 17 RCTs were summarized according to subject, study design, duration and dosage of supplementation, protocol of exercise, time-points of measurement, and supplementation effects on biomarkers (Table 1). The age of subjects of selected studies were within the range of 18 to 40 years. Vitamin E dosage was standardized to IU format (1 mg = 1.21 IU) ranging from 300 to 1318 IU per day. Measurement time-points were varied from immediately after exercise to 7 days after exercise. Four studies were conducted on athlete and 13 studies on heathy people. Gender of subjects was almost all male in selected RCTs. There was one RCT conducted on both sexes and the other RCT conducted on only females. Effect of vitamin E supplementation was analyzed on muscle damage measured by CK and LDH, on oxidative stress demonstrated by MDA and TAS, and on inflammation, represented by IL-6.

### 3.3. Quality Analysis of RCT Included in the Meta-Analysis

Quality assessment of 17 RCTs included in the meta-analysis was performed by Cochrane collaboration risk of bias tool [27] (Figure 2). Nine RCTs were evaluated ‘low risk’ of bias and 8 RCTs were considered as ‘moderate risk’ of bias.

### 3.4. Effects of Dietary Vitamin E Supplementation on Exercise-Induced Muscle Damage

CK and LDH concentration were used to examine the effect of dietary vitamin E supplementation on exercise-induced muscle damage. CK concentration (U/L) was analyzed in 10 RCTs. The overall effect of vitamin E supplementation on muscle damage was significant (SMD −1.00, 95% CI: 1.95 to −0.06, *p* = 0.04) with high heterogeneity (I^2^ = 90%) (Figure 3A). Subgroup analyses were conducted according to measurement time-points, vitamin E dosage, and type of subject (Table 2). Vitamin E supplementation had a significant effect on muscle damage immediately after exercise (SMD −1.89, 95% CI: −3.39 to −0.39, *p* = 0.01), but no protective effect at 24 h (SMD −0.084, 95% CI: −2.31 to 0.63, *p* = 0.26) and 48 h after exercise (SMD 0.71, 95% CI: −2.43 to 3.86, *p* = 0.66). Heterogeneity was high at immediately, 24 h, or 48 h after exercise (I^2^ = 91%, 88%, 94%, respectively). Low dosage of vitamin E supplementation (≤500 IU per daily) showed ameliorative effect on muscle damage (SMD −1.94, 95% CI: −2.99 to −0.89, *p* = 0.0003) with high heterogeneity (I^2^ = 88%). However, high doses of vitamin E supplementation (> 500 IU per day) had no effect on muscle damage (SMD 0.73, 95% CI: 1.27 to 2.73, *p* = 0.48) with high heterogeneity (I^2^ = 92%). Moreover, vitamin E supplementation significantly decreased CK concentration (SMD −5.15, 95% CI: −9.92 to −0.39, *p* = 0.03) in athletes, but there was no beneficial effect in non-athletes (SMD −0.31, 95% CI: −1.21 to 0.58, *p* = 0.49) with high heterogeneity (I^2^ = 93, 88% respectively). Egger’s test (*p* = 0.45) showed no significant publication bias related to the association between vitamin E supplementation and CK. 

Another muscle damage maker, LDH concentration (U/L) was analyzed in 4 RCTs. Vitamin E supplementation showed a significant reduction of LDH immediately after exercise (SMD −1.80, 95% CI: −3.21 to −0.39, *p* = 0.01) with high heterogeneity (I^2^ = 82%) (Figure 3B).

### 3.5. Effects of Dietary Vitamin E Supplementation on Exercise Induced Oxidative Stress

Exercise-induced oxidative stress was examined using plasma MDA and TAS concentrations to examine the effect of dietary vitamin E supplementation. MDA concentration (μmol/L) was analyzed in seven RCTs. The overall effect of vitamin E supplementation resulted in no protective effect on MDA (SMD −0.17, 95% CI: −0.52 to 0.18, *p* = 0.35), with low heterogeneity (I^2^ = 42%) (Figure 4A). Subgroup analysis was conducted according to measurement time-points, vitamin E dosage and a type of subject (Table 3). Vitamin E supplementation did not have a significant effect on exercise-induced oxidative stress immediately after exercise (SMD −0.21, 95% CI: −0.57 to 0.14, *p* = 0.24), at 24 h (SMD 0.20, 95% CI: −0.41 to 0.81, *p* = 0.52) and 48 h after exercise (SMD −0.92, 95% CI: −2.23 to 0.39, *p* = 0.17) with no, low and moderate heterogeneity, respectively (I^2^ = 0%, 37%, 76%). Lower dosage of vitamin E supplementation (≤500 IU) had beneficial effect on oxidative stress (SMD −0.48, 95% CI: −0.95 to −0.01, *p* = 0.04) with no heterogeneity (I^2^ = 0%). High dosage of vitamin E supplementation (>500 IU per daily) had no significant effect on oxidative stress (SMD −0.06, 95% CI: −0.49 to 0.38, *p* = 0.79) with low heterogeneity (I^2^ = 47%). Vitamin E supplementation in both athletes (SMD −0.55, 95% CI: −2.00 to 0.90, *p* = 0.46) and non-athletes had no potential benefit on MDA concentration (SMD −0.15, 95% CI: −0.43 to 0.13, *p* = 0.30) with high and no heterogeneity, respectively (I^2^ = 80%, 0%). Egger’s test (*p* = 0.89) showed no publication bias related to the association between vitamin E supplementation and exercise-induced MDA level. 

Another oxidative stress maker, TAS concentration (nmol/L) was analyzed in two RCTs. Vitamin E supplementation significantly reduced TAS level immediately after exercise (SMD −0.81, 95% CI: −1.46 to −0.16, *p* = 0.01) with no heterogeneity (I^2^ = 0%) (Figure 4B).

### 3.6. Effect of Dietary Vitamin E Supplementation on Exercise-Induced Inflammation

IL-6 concentration was used to examine the effect of dietary vitamin E supplementation on exercise-induced inflammation in this meta-analysis. IL-6 concentration (pg/mL) was analyzed in two RCTs. Vitamin E supplementation did not have a protective effect on inflammation after exercise (SMD −0.39, 95% CI: −4.85 to 4.12, *p* = 0.087). There was moderate heterogeneity between two studies (I^2^ = 57%) (Figure 5).

## 4. Discussion

This study investigated the effects of dietary vitamin E supplementation on exercise-induced muscle damage, oxidative stress, and inflammation. Our results demonstrate the protective role of vitamin E supplementation in detrimental outcome in muscle due to exercise. Interestingly, beneficial effects of vitamin E supplementation on muscle damage showed singularity of optimal dosage or had different pattern between athletes and non-athletes.

Various types of exercise generate ROS, and exaggerated ROS after exercise may lead to redox imbalance states, commonly known as oxidative stress and chronic inflammation [7]. Oxidative stress and inflammation accelerated muscle fatigue, delayed recovery time and reduced exercise performance [9]. Vitamin E was suggested as a nutritional strategy for preventing harmful effects of exercise [11]. However, some research showed that antioxidants had no effects on exercise-induced muscle damage, oxidative stress, and inflammation [6]. As a result, it was at the center of a controversy whether antioxidants have beneficial effects against intensive exercise. Hence, the present study investigated the effects of dietary vitamin E supplementation on exercise-induced muscle damage, oxidative stress, and inflammation. 

Muscle damage is related to transient ultrastructural myofibrillar disruption, which increases efflux of myocellular enzymes such as CK and LDH [36]. Dietary vitamin E supplementation more evidently decreased exercise-induced muscle damage immediately after exercise, demonstrated by CK. Given that CK concentrations typically peak within 24 h of exercise [36], the results suggested that vitamin E supplementation protected muscle damage more effectively at an early stage. Interestingly, protective effects of vitamin E supplementation on exercise-induced muscle damage was relatively clear with lower dosage (≤500 IU) and athletes. When compared to non-athletes, athletes who exercise on a regular basis have stronger antioxidant defense systems and lower oxidative stress [37]. This implies that vitamin E supplementation might work better at a certain dosage (probably at 500 IU) or propel a well-constructed antioxidant defense system. Even though the number of studies included was small, vitamin E supplementation decreased exercise-induced LDH concentration which is another muscle damage marker. Therefore, additional studies are needed to assess whether vitamin E supplementation can downgrade exercise-induced LDH concentration.

Oxidative stress results in oxidation of cellular lipids, typically referred to as lipid peroxidation, that produces various oxidative products such as MDA [13]. Overall, vitamin E supplementation generally had no effect on MDA concentration; however, again, 500 IU or less vitamin E did reduce MDA concentration immediately after exercise when MDA products are typically at their peak [38]. Two studies illustrated that vitamin E supplementation reduced exercise-induced TAS concentration. Pre- and post-exercise TAS concentrations were higher in the vitamin E supplemented group than in the placebo, but TAS levels were more significantly reduced after exercise in the vitamin E supplemented group [15,32]. It could be interpreted that vitamin E supplementation increased the antioxidant capacity before exercise and inhibited the free radical production during exercise. 

In damaged muscle cells, exercise-induced inflammation results in the secretion of pro-inflammatory cytokines such as IL-6 and TNF-a [39]. In this study, vitamin E supplementation did not reduce exercise-induced IL-6 concentration immediately after exercise. However, after screening, only two studies with small numbers of participants were available to analyze exercise-induced change in IL-6 as a parameter of inflammation. Therefore, additional studies are needed to assess the effects of vitamin E supplementation on exercise-induced IL-6.

The current meta-analysis found that vitamin E had a protective effect on exercise-induced muscle damage and oxidative stress, especially when measured immediately after exercise, in athletes, and with 500 IU or less vitamin E supplementation. It is noteworthy that a previous meta-analysis reported that total mortality increased with a higher dosage of vitamin E supplementation [40]. As a result, people who consumed low doses of vitamin E supplementation (500 IU) and exercised were expected to have beneficial effects on exercise-induced muscle damage and oxidative stress.

Even though there have been lots of meta-analyses for understanding the impact of vitamin E supplementation on exercise-induced muscle damage, the current study provides distinct perspectives on optimal vitamin E supplementation. First, the current study considered the dosage effect of vitamin E supplementation. The daily dosage of vitamin E supplementation used in the present study was divided into high and low dosages based on the median value (500 IU) and the effects were analyzed on exercise-induced muscle damage (CK) and oxidative stress (MDA). The results in this study showed that lower dosage of vitamin E supplementation (≤500 IU) had greater protective effects. Finally, our results showed a complementary relationship among exercise-induced muscle damage, oxidative stress, and inflammation by confirming the results at once. 

Nevertheless, there were also some limitations in the present study. First, other markers such as PC, 8-OHdG, and TNF- α could not be analyzed due to an insufficient number of studies, especially using vitamin E alone as a supplement. Although particular markers were extensively analyzed, additional studies should be performed to analyze more biomarkers to achieve solid results. Second, there are many differences among the studies used in this meta-analysis. Despite of subgroup analysis of time-point, daily dose, and type of subject, differences in exercise types and intensities and duration of supplementation could be attributed to high heterogeneity and discrepancy of results among the studies.

## 5. Conclusions

In conclusion, this meta-analysis showed that dietary vitamin E supplementation significantly reduced biomarkers related to exercise-induced muscle damage and oxidative stress. The protective effect of vitamin E became apparent when CK and MDA were measured immediately after exercise or in athletes. Furthermore, low doses of vitamin E supplementation (≤500 IU/day) had significant protective effects against exercise-induced muscle damage and oxidative stress. However, vitamin E supplementation was not significantly effective on exercise-induced inflammation, despite the small number of RCTs included. More RCTs are required to analyze various factors generating high heterogeneity to clarify the effects of dietary vitamin E supplementation on exercise-induced muscle damage, oxidative stress, and inflammation.

## Figures and Tables

**Figure 1 nutrients-14-01599-f001:**
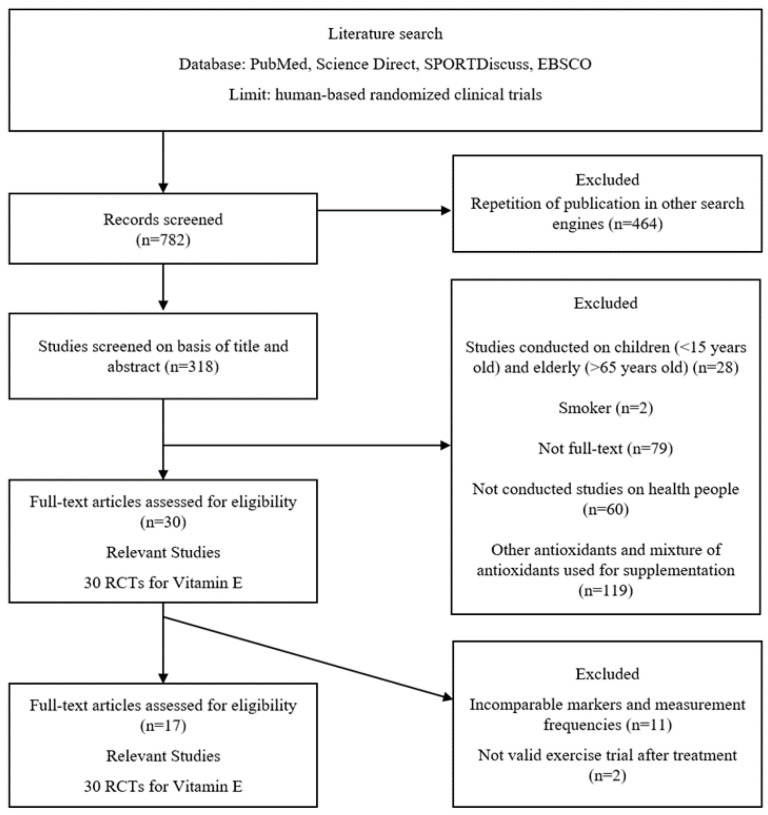
Flow chart illustrating the study selection process.

**Figure 2 nutrients-14-01599-f002:**
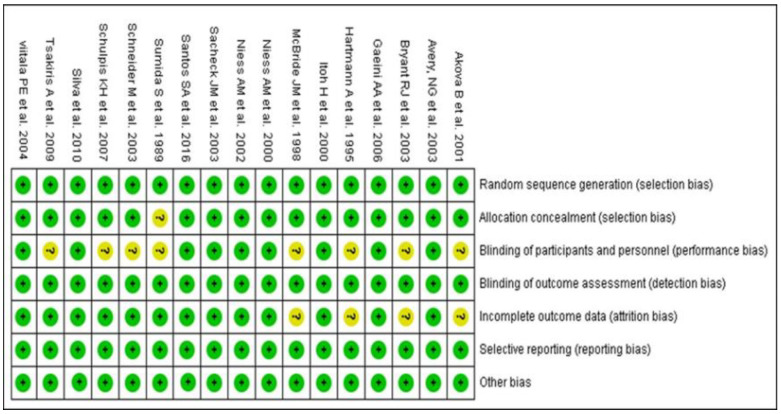
Quality analysis of RCTs included in the meta-analysis (+, ‘yes’; ?, ‘unclear’).

**Figure 3 nutrients-14-01599-f003:**
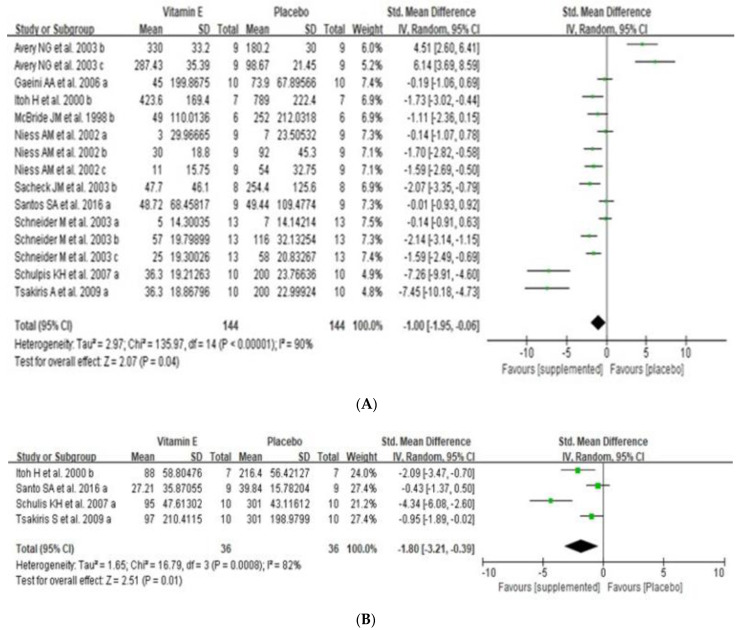
Forest plot for overall effects of vitamin E supplementation on CK (**A**) and LDH (**B**) CK, creatine kinase; LDH, lactate dehydrogenase; SD, standard deviation; SE, standard error; CI, confidence interval; df, degrees of freedom; a, measurement immediately after exercise; b, measurement at 24 h after exercise; c, measurement at 48 h after exercise. Green squares represent the weighted mean difference (WMD) of each study and the black diamond represents the summary of weight mean difference.

**Figure 4 nutrients-14-01599-f004:**
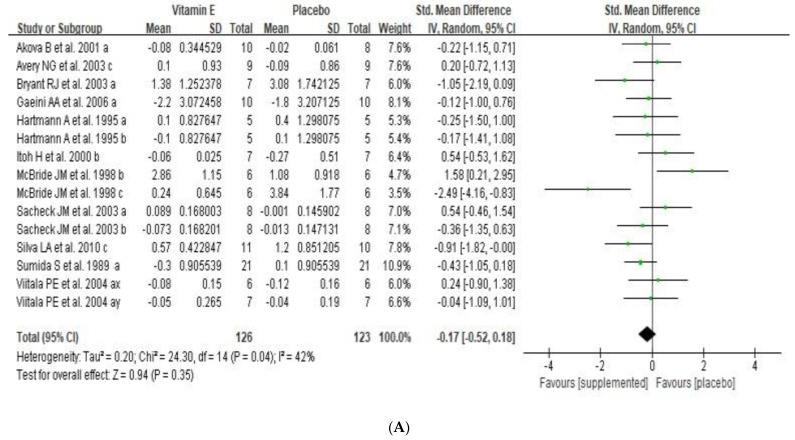
Forest plot for overall effects of vitamin E supplementation on MDA (**A**) and TAS (**B**) MDA, malondialdehydes; TAS, total antioxidant status; SD, standard deviation; SE, standard error; CI, confidence interval; df, degrees of freedom; a, measurement immediately after exercise; b, measurement at 24 h after exercise; c, measurement at 48 h after exercise; x, pre-training exercise test; y, post-training exercise test. Green squares represent the weighted mean difference (WMD) of each study and the black diamond represents the summary of weight mean difference.

**Figure 5 nutrients-14-01599-f005:**
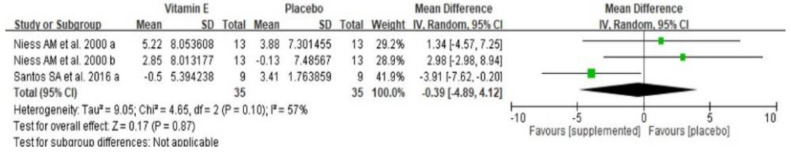
Forest plot for an overall effect of vitamin E supplementation on IL-6 (interleukin-6); SD, standard deviation; CI, confidence interval; df, degrees of freedom; a, measurement immediately after exercise; b, measurement at 24 h after exercise. Green squares represent the weighted mean difference (WMD) of each study and the black diamond represents the summary of weight mean difference.

**Table 1 nutrients-14-01599-t001:** Characteristics of included studies for the meta-analysis of effects of dietary vitamin E supplementation.

Study	Year	Study	Supplementation		Subjects	Exercise Protocol	Measurement	Muscle Damage		Oxidative Stress		Inflammation	
(First Author)		Design	Duration (day)	Daily Dosage (IU)	(Number, Age)			Marker	Effect	Marker	Effect	Marker	Effect
Santos S.A. [23]	2016	RCT, double-blind design	Single dose (1)	302.5	9, healthy males, 24.2 ± 2.2 years	60 min at an intensity of 70% VT I in normoxia and hypoxia simulating an altitude of 4200 m	before and immediately, at 1 h after exercise	CK LDH	↓ ↓			IL-6	↓
Schulpis K.H. [26]	2007	RCT	30	300	10, male basketball players, 18.5 ± 0.6 years	Stretching, technical-tactical part, a heavy training load part	before, immediately after the training	CK LDH	↓ ↓	TAS	↓		
Gaeini A.A. [21]	2006	RCT, double-blind design	42	672	20, male students, 23.1 ± 2.0 years	incremental exercise test	before and immediately after exercise	CK	↔	MDA	↔		
Avery N.G. [22]	2003	RCT, double-blind design	31	1200	18, healthy men, 22.7 ± 4.1 years	repeated bouts of whole-body resistance exercise	each day beginning with the first exercise session at, 24 h and 48 h after exercise	CK	↑	MDA	↔		
Sacheck J.M. [30]	2003	RCT, double-blind design	84	1000	16, healthy men, 26.4 ± 3.3 years	ran downhill for 45 min at 75% VO_2_ max	before (baseline) and immediately after exercise (0 h), and at 6, 24, and 72 h after exercise	CK	↓	MDA	↔		
Itoh H. [31]	2000	RCT, double-blind design	28	1200	14, male students, 21.1 ± 2.3 years	6-day running training session (48.3 ± 5.7 km × day^–1^)	baseline, the day immediately before, the next day after, and three weeks after the 6 day running training	CK	↓	MDA	↓		
Mcbride J.M. [17]	1998	RCT	14	1200	12, weight-trained males VE: 22.2 ± 0.7 years P: 22.0 ± 0.9 years	heavy resistance exercise	before and at 24 h and 48 h after exercise	CK	↓	MDA	↑		
Bryant R.J. [18]	2003	RCT,	21	400	7, male cyclists, 22.3 ± 2 years	60 min steady state ride and a 30-min performance ride	before and immediately after exercise			MDA	↓		
Viitala P.E. [32]	2004	RCT, double-blind, crossover design	14	1318	27, males and females, ages of 19 and 30 years	resistance exercise test	before and immediately and 6 h after exercise			MDA	↔		
Niess A.M. [16]	2002	RCT, double-blind, crossover design	8	500	9, healthy males, 25.3 ± 1.0 years	incremental exercise test + continuous run	before the beginning of supplementation and 3, 24 and 48 h after the end of the continuous run	CK	↓				
Schneider M. [33]	2003	RCT, crossover design	8	500	13, males, 26.5 ± 0.9 years	incremental exercise test + continuous run	at rest, 0, 0.25, 1, 3, 24 and 48 h after exercise.	CK	↔				
Tsakiris S. [15]	2009	RCT	30	300	10, male basketball players, 18.5 ± 0.6 years	training program two or three times a week	before, immediately after the training	CK LDH	↓ ↓	TAS	↓		
Hartmann A. [34]	1995	RCT	14	1200	8, men, 29–34 years	a single bout of exhaustive exercise	before and at 15 min and 24 h after exercise			MDA	↓		
Sumida [20]	1989	RCT	28	447	21, healthy male college students, 20.3 ± 0.3 years	incremental exercise test	before, immediately after exhaustion, and at 1 and 3 h in the recovery period.			MDA	↓		
Silva L.A. [19]	2010	RCT, double-blind design	14	800	21, male volunteers, 22.5 ± 4 years	EE	days 0, 2, 4, and 7 after EE			MDA	↓		
Akova B. [35]	2001	RCT	56	300	18, sedentary women, 19–35 years	fatigue test	before and after the exercise			MDA	↔		
Niess A.M. [24]	2000	RCT, double-blind design	56	500	38, triathletes, VE: 35.2 ± 1.6 years P: 39.2 ± 1.4 years	the race included a 3.9-km ocean swim, 180-km bike race and 42-km run	before the race, at 0 h, 3 h, 24 h, and 48 h postrace					IL-6	↑

Abbreviations: CK, creatine kinase; LDH, lactate dehydrogenase; MDA, malondialdehyde; TAS, total antioxidant status; IL-6, interleukin-6; IL-10, interleukin-10; RCT, randomized controlled trial; VT, ventilatory threshold; EE, eccentric exercise; VE, Vitamin E supplementation; P, Placebo; ↓, decrease; ↑, increase; ↔, no difference.

**Table 2 nutrients-14-01599-t002:** Results of subgroup analysis for the effect of vitamin E supplementation on CK concentration.

Group	No. of Subject	Std, Mean Difference in CK, U/L (95%CI)	*p*	P_heterogeneity_	I^2^ (%)
**Total**	288	−1.00, (−1.95 to −0.06)	0.04	<0.00001	90
**Measurement timepoint**					
Immediately after exercise	122	−1.89, (−3.39 to −0.39)	0.01	<0.00001	91
at 24 h after exercise	104	−0.084, (−2.31 to 0.63)	0.26	<0.00001	88
at 48 h after exercise	61	0.71, (−2.43 to 3.86)	0.66	<0.00001	94
**Daily dosage**					
≤500	190	−1.94, (−2.99 to −0.89)	0.0003	<0.00001	88
>500	98	0.73, (−1.27 to 2.73)	0.48	<0.00001	92
**Subject**					
Athlete	52	−5.15, (−9.92 to −0.39)	0.03	<0.00001	93
Non-athlete	236	−0.31, (−1.21 to 0.58)	0.49	<0.00001	88

**Table 3 nutrients-14-01599-t003:** Results of subgroup analysis for effects of vitamin E supplementation on MDA concentration.

Group	No. of Subject	Std, Mean Difference in CK, U/L (95%CI)	*p*	P_heterogeneity_	I^2^ (%)
**Total**	249	−0.17, (−0.52 to 0.18)	0.35	0.04	42
**Measurement timepoint**					
Immediately after exercise	126	−0.21, (−0.57 to 0.14)	0.24	0.71	0
at 24 h after exercise	72	0.20, (−0.41 to 0.81)	0.52	0.36	37
at 48 h after exercise	51	−0.92, (−2.23 to 0.39)	0.17	0.04	76
**Daily dosage**					
≤500	74	−0.48, (−0.95 to −0.01)	0.04	0.53	0
>500	175	−0.06, (−0.49 to 0.38)	0.62	0.04	47
**Subject**					
Athlete	38	−0.55, (−2.00 to 0.90)	0.46	0.002	80
Non-athlete	211	−0.15, (−0.43 to 0.13)	0.30	0.68	0

## Data Availability

Data are available by request to the authors.

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
