# Peer review of "Can Low-Dose of Dietary Vitamin E Supplementation Reduce Exercise-Induced Muscle Damage and Oxidative Stress? A Meta-Analysis of Randomized Controlled Trials"

_nutrients, 2022, doi:10.3390/nu14081599_

Round 1

Reviewer 1 Report

The manuscript presents an interesting topic. However, in my opinion, there are many issues.

  1. The meta-analysis including similar articles has been published yet.   https://doi.org/10.1139/apnm-2013-0566
  2. Authors included articles published from 1989 to 2016, it is a very wide range of time, and some of them have over 30 years! with the different analytical methods, the lifestyle of participants, etc.
  3. It is worth mentioning that in included studies usually athlete subjects take a low dose of vitamin E and this may have a significant influence on the results of this is meta-analysis.
  4. Results are related to young people 18-30 years and this should be included in the conclusion.
  5. Parameter of inflammation (IL-6)  was analyzed only in two studies on rather a small number of participants, and one of the studies included 38 triathletes and the other 9 healthy males, so these two studies were not comparable.
  6. Discussion is rather poor writing. I can not find the novelty of this study and in my opinion, the results should be re-discussed, interactions between factors should be taken into account, and also it is important that results should be based on rather current studies.

Author Response

Dear Editor and All Reviewers,

We would like to express our gratitude for your fruitful comments and suggestions. We believe your feedback has led to an improved and strengthened manuscript. The corrections and changes of the manuscript are explained point by point in this response letter. In the revised manuscript, all edits are tracked. Again, we appreciate your critical reviews.

Reviewer comments and our response:

Reviewer 1

The manuscript presents an interesting topic. However, in my opinion, there are many issues.

  1. The meta-analysis including similar articles has been published yet. https://doi.org/10.1139/apnm-2013-0566

The article you mentioned did not find significant protection against either exercise-induced lipid peroxidation (measured by MDA) or muscle damage (examined by CK). However, there are two main big gaps between the article and our current study as follows:

First, since the range of dosage of vitamin E supplementation in each study was wide, we conducted this meta-analysis by dividing the vitamin E supplementation dosage (≤500, >500 IU) which was one of the key findings in our study. The results of this meta-analysis demonstrated that vitamin E supplementation less than 500 IU had protective effects against exercise induced muscle damage whereas vitamin E supplementation higher than 500 IU had not effects.

Moreover, this study included various markers such as not only CK and MDA but also IL-6, TAS, and LDH which are markers of muscle damage and inflammation. Additional markers we analyzed allowed multilateral examination on the effects of low-dose of dietary vitamin E supplementation on exercise-induced muscle damage and oxidative stress/inflammation.

For these reasons, the current study has novelty in the field of vitamin E research related to exercise and considered multiple aspects of impact from vitamin E supplementation on muscle damage induced by exercise. Furthermore, effect of vitamin E supplementation on exercise-induced muscle damage was affected by a certain concentration of vitamin E supplementation.

  1. Authors included articles published from 1989 to 2016, it is a very wide range of time, and some of them have over 30 years! with the different analytical methods, the lifestyle of participants, etc.

Thank you for your comment. Even though differences between past studies and recent studies may exist, meta-analysis is often adopted as a method to integrates the results of multiple scientific researches for long periods of time. For this reason, meta-analysis can generate quantitative estimate of the effectiveness of the intervention with little influence from study years conducted or demographic background of subjects. In addition, there were the limited number of studies of vitamin E intervention study without mixing other antioxidants such as vitamin C in the recent years. Thus, our group included study as many as possible to provide integrative and objective interpretation on vitamin E supplementation for exercise-induced muscle damage.

  1. It is worth mentioning that in included studies usually athlete subjects take a low dose of vitamin E and this may have a significant influence on the results of this is meta-analysis.

As shown in Table 1, athlete not always took a low dose of vitamin E such as Mcbride et al. (1998). In addition, when we analyzed the effect of vitamin E supplementation on each biomarker, we divided groups by daily dosage OR athletes/non-athletes which did not considered interaction between dosage and subject. Daily dosage and subject are different factor in each analysis. To clarify this, we edited the Discussion section clearly.

  1. Results are related to young people 18-30 years and this should be included in the conclusion.

Thank you for your comment. The manuscript was revised according to your suggestion (Line 181~182). To be specific, the age of subjects of selective studies were within the range of 18 to 40 years.

  1. Parameter of inflammation (IL-6) was analyzed only in two studies on rather a small number of participants, and one of the studies included 38 triathletes and the other 9 healthy males, so these two studies were not comparable.

We had recognized that IL-6 was not reasonable to be analyzed as a parameter of inflammation due to small number of researches. Also, we had questions about the comparability of the included studies. However, we expected that analyzing the IL-6 index itself would be meaningful to other researchers and wanted to inform that additional studies are in need to reach the result even though there were only two studies to analyze. This limitation was described in the Discussion section of revised manuscript as well.

  1. Discussion is rather poor writing. I cannot find the novelty of this study and in my opinion, the results should be re-discussed, interactions between factors should be considered, and also it is important that results should be based on rather current studies.

As you responded to question A, our group emphasized the singularity of vitamin E supplementation in terms of dosage (500 IU). This is the main difference between the previous meta-analysis on effects of vitamin E in exercise-induced muscle damage. In addition, we revised discussion to elaborate novelty and meaning of the current study. Again, thank you for all your fruitful comments to make our paper better. 

Reviewer 2 Report

I think the topic of study is of interest
The intro is interesting
Current sources consulted
Carrying out bibliographic reviews and meta-analyses I think is very timely and requires very specific and laborious techniques.
Some considerations:
In the abstract I think it should be clear that it is a meta-analysis
Also in the abstract you should make the procedure clear. At least one review on the analysis strategies, how these were carried out
In the method and results section, it would have been interesting to explain the procedure. Whether or why the PRISMA procedure is used.
The results section seems appropriate
I also find the discussion section interesting.

Author Response

Dear Editor and All Reviewers,

We would like to express our gratitude for your fruitful comments and suggestions. We believe your feedback has led to an improved and strengthened manuscript. The corrections and changes of the manuscript are explained point by point in this response letter. In the revised manuscript, all edits are tracked. Again, we appreciate your critical reviews.

Reviewer 2

I think the topic of study is of interest. The intro is interesting. Current sources consulted. Carrying out bibliographic reviews and meta-analyses I think is very timely and requires very specific and laborious techniques. Some considerations:

  1. In the abstract I think it should be clear that it is a meta-analysis

The abstract was edited according to your suggestion (Line 17).

  1. Also in the abstract you should make the procedure clear. At least one review on the analysis strategies, how these were carried out

Thank you for your comments. We clarified the procedure in the abstract as you mentioned. (Line 21~22)

  1. In the method and results section, it would have been interesting to explain the procedure. Whether or why the PRISMA procedure is used.

The current meta-analysis selected studies by following PRISMA procedure which is widely used. The statement related to this was added in the Method section. (Line126~128)

  1. The results section seems appropriate. I also find the discussion section interesting.

We appreciate for your comments to make our paper better.
